# GazeVQA: A Video Question Answering Dataset for Multiview Eye-Gaze Task-Oriented Collaborations

**Muhammet Furkan Ilaslan**[1,2*]    **Chenan Song**[1*]    **Joya Chen**[1]    **Difei Gao**[1]
**Weixian Lei**[1]    **Qianli Xu**[2†]    **Joo Hwee Lim**[2]    **Mike Zheng Shou**[1]

[1]Show Lab, National University of Singapore

[2]Institute for Infocomm Research, Agency for Science, Technology, and Research (A*STAR), Singapore

{m.furkanilaslan, chenan.song}@u.nus.edu , mike.zheng.shou@gmail.com

{stuimf, qxu, joohwee}@i2r.a-star.edu.sg

## Abstract

The usage of exocentric and egocentric videos in Video Question Answering (VQA) is a new endeavor in human-robot interaction and collaboration studies. Particularly for egocentric videos, one may leverage eye-gaze information to understand human intentions during the task. In this paper, we build a novel task-oriented VQA dataset, called GazeVQA, for collaborative tasks where gaze information is captured during the task process. GazeVQA is designed with a novel QA format that covers thirteen different reasoning types to capture multiple aspects of task information and user intent. For each participant, GazeVQA consists of more than 1,100 textual questions and more than 500 labeled images that were annotated with the assistance of the Segment Anything Model. In total, 2,967 video clips, 12,491 labeled images, and 25,040 questions from 22 participants were included in the dataset. Additionally, inspired by the assisting models and common ground theory for industrial task collaboration, we propose a new AI model called AssistGaze that is designed to answer the questions with three different answer types, namely textual, image, and video. AssistGaze can effectively ground the perceptual input into semantic information while reducing ambiguities. We conduct comprehensive experiments to demonstrate the challenges of GazeVQA[1] and the effectiveness of AssistGaze[2].

## 1  Introduction

Computer vision systems for understanding data, such as video clips, photos, and generated synthetic images, have been studied for a few decades and

---

*Equal contribution.

†Corresponding author.

[1]https://github.com/mfurkanilaslan/GazeVQA

[2]https://github.com/showlab/AssistGaze

**Question:** Which component will be assembled next?

**Textual Answer** (Multiple Choice)

A1: Hub cover bolts
A2: Casing bolts
A3: Input hub cover
A4: Output hub cover

**Image Answer**

**Video Answer** (Step2-6 video clip was used)  t1: 00: 10 – t2: 00:30 sec

TPV-enface

TPV-sideward

FPV-raw

FPV-gaze

Figure 1: GazeVQA dataset contains gaze info with ego/exocentric videos to answer the questions that have semantic knowledge about human-robot studies.

have achieved remarkable progress (Deng et al., 2009; He et al., 2016; Russell et al., 2008; Feichtenhofer et al., 2019; Tapaswi et al., 2016; Krizhevsky et al., 2017; Yu et al., 2019). Many projects (Wu et al., 2017; Gao et al., 2021; Sood et al., 2021; Antol et al., 2015; Kafle and Kanan, 2017; Lei et al., 2022; Gao et al., 2022) in recent years have shown promising developments in the usage of multimodal information. Moreover, there are several VQA studies, such as TutorialVQA (Colas et al., 2020), AGQA (Grunde-McLaughlin et al., 2021), TVQA (Lei et al., 2018), KnowIT VQA (Garcia et al., 2020), EgoVQA (Fan, 2019), Ego-TaskQA (Jia et al., 2022), GQA (Hudson and Man-

ning, 2019), DramaQA (Choi et al., 2021), VQA-MHUG (Sood et al., 2021), PointQA (Mani et al., 2020) that target at different aspects such as instructional assistance, subtitle inclusion, and compositional reasoning. Despite the notable progress in VQA studies, most of the works use videos from one or multiple third-person views (TPV). Constrained by the fixed viewing angle, TPV video may not be able to capture subtle information on human operation in industrial tasks such as assembly and disassembly. In particular, the TPV may not be sufficient to capture the interaction that forms the basis of human-human/robot shared tasks. First-person view (FVP) videos containing gaze information are beneficial to understanding the tasks performed in assembly-disassembly studies and overcoming spatial ambiguities.

A novel task-oriented collaborative VQA is proposed in this research based on the collaboration theory, which is defined as a coordinated, synchronous activity resulting from a continuous effort to construct and maintain a shared understanding of a problem (Dillenbourg et al., 1996; Roschelle and Teasley, 1995). This theory encompasses four key aspects: *situation*, *interaction*, *process* (grounding), and *effect* (Dillenbourg et al., 1996). *Situation* is characterized by a degree of division of labor, which sets the stage for interactions. *Interaction* is characterized by cognitive *process* such as grounding and mutual modeling, leading to *effect* that contribute to shared understanding (Dillenbourg et al., 1996). The ultimate goal of collaboration, according to this theory, is to build the common grounds necessary for effective cooperation. In the context of interaction between two subjects, there is a requirement to develop a shared language for task completion. This concept is called grounding mechanism (Dillenbourg et al., 1996), which operates on a functional principle: while perfect mutual understanding is not attainable, nor necessary, it is required to sustain sufficient mutual understanding to continue the task (Clark and Brennan, 1991). In addition, every collaboration has a unique user experience. However, the TPV FPV videos may not adequately capture the semantic meaning that represents users' intentions and unique experiences.

The GazeVQA task aims to establish common ground for collaborative tasks involving human-robot interaction. Furthermore, different from other existing VQA studies, GazeVQA leverages common grounding theory (Li et al., 2006) and uses eye-gaze information to facilitate semantic understanding in collaboration tasks. As shown in Figure 1, GazeVQA contains gaze-augmented QA pairs that are related to human-human interaction, which can assist a person in learning task knowledge with an AI assistant that answers questions.

To tackle the challenges of visual understanding presented in GazeVQA, we propose AssistGaze -an assistive model for task-collaborative HRI-QA. It is a combination of different models to answer the questions with three different answer types, including textual answers, video retrieval answers, and image answers. The main purpose of AssistGaze is to predict correct answers for task-specific questions in GazeVQA by using gaze information. We inquire into different configurations of the proposed model, such as availability of gaze, choice of encoders, answer types, etc., and compare our model with a few benchmarks, such as random guess and VIOLET (Fu et al., 2021). First, random guess models and masked versions of all video features are examined. Secondly, the effectiveness of TPV videos is examined. Finally, FPV videos with gaze information are added to the experiments. It is observed that the model obtains the highest performance when all features are included.

There are three contributions of this paper. (1) We propose a novel collaborative VQA task to support collaboration in industrial applications such as the assembly and disassembly processes. (2) We design GazeVQA, a new VQA dataset on collaborative scenarios, with carefully curated QA pairs and multimodal inputs including videos captured from multiviews and eye gaze information in the FPV video. (3) We create a new assistant model, AssistGaze, to generate answers in three formats, while using gaze information to resolve spatial ambiguities. The model assists a human user in collaboration and interaction by answering task-specific questions.

## 2 Related Works

### 2.1 Video QA

There is increasing momentum of VQA studies recently (Antol et al., 2015; Lu et al., 2019; Goyal et al., 2017; Gao et al., 2020; Tapaswi et al., 2016; Maharaj et al., 2017; Mun et al., 2017; Jang et al., 2017; Gao et al., 2023; Jang et al., 2017). With the acceleration of the studies carried out in this field, human-centered action recognition studies were conducted with video clips taken from TV

| Dataset | Total Hours | # of Videos | Avg. L. (min.) | # of Verbs | # of Actions | # of Objects |
|---|---|---|---|---|---|---|
| Ego4D (Grauman et al., 2022) | 120 | - | - | 74 | 87 | - |
| MECCANO (Ragusa et al., 2021) | 6.9 | 20 | 20.7 | 12 | 21 | 61 |
| IKEA ASM (Ben-Shabat et al., 2021) | 35 | 371 | 5.6 | 12 | 10 | 33 |
| Assembly101 (Sener et al., 2022) | 513 | 4,321 | 7.1 | 24 | 90 | 1,380 |
| EPIC-KITCHENS (Damen et al., 2021) | 100 | 700 | 8.5 | 97 | 300 | 4,053 |
| **GazeVQA(gaze)** | 31.5 | 895 | 2.5 | 28 | 16 | 40 |
| **GazeVQA** | 125 | 2,967 | 2.5 | 28 | 16 | 40 |

Table 1: Comparison of the GazeVQA and other action datasets in the context of total hours of video clips, number of videos, average length of video clips (in minutes), number of verbs, actions, and objects.

series (Lei et al., 2020). A detailed comparison of GazeVQA and current VQA models is in the appendix, Tables 8 and 9. Egocentric videos have become popular in the computer vision community due to the prevalence of small cameras and the ease of collecting FPV videos. These cameras are useful to collect a variety of videos in different domains, such as manufacturing (assembly-disassembly) (Tan et al., 2020), education, behavior (Bambach et al., 2017; Wong et al., 2022), and sports (Bertasius et al., 2017; Shi and Bertasius, 2017). These videos also enable analysis such as object detection (Fathi et al., 2011; Fan et al., 2018; Furnari et al., 2017), hands detection (Bambach et al., 2015; Chen et al., 2023), gaze detection and prediction (Huang et al., 2018).

To train powerful VQA models, datasets with reasonable scale and accurate annotation are necessary. Large-scale datasets such as EPIC-KITCHENS (Damen et al., 2021) or Ego4D (Grauman et al., 2022) are not suitable for annotation due to their enormous sizes. Other studies for industrial assembly-disassembly are Assembly101 (Sener et al., 2022), IKEA ASM (Ben-Shabat et al., 2021), and MECCANO (Ragusa et al., 2021). Table 1 presents a comparison of various works alongside GazeVQA. GazeVQA is the first dataset that deals with assembly-disassembly in the industry context that contains eye gaze information. The availability of eye gaze information is a unique feature of our dataset, which can augment semantic understanding of the tasks by harnessing human intentions. Compared to MECCANO and IKEA ASM datasets, GazeVQA is superior in terms of total recording hours, number of video clips, and number of different action words. Consequently, it supports model development towards real-world applications. Interested readers may refer to Tables 8 and 9 in the appendix for more comprehensive comparisons between GazeVQA and existing VQA datasets.

## 2.2 FPV and gaze information

Visual grounding is a recent research topic in the VQA studies. (Zhu et al., 2016; Hudson and Manning, 2019) proposes a method to localize objects to answer the questions with visual grounding. However, human-robot interaction requires more cost-effective solutions. Thus, the usage of hand gestures, i.e. pointing at objects, shows a useful approach (Mani et al., 2020), which nevertheless, suffers from a tedious process for recording hand actions and visual processing. Alternatively, human gaze can be used to decode the human intention with respect to localizing objects and recognizing actions.

Gaze information has been used for action recognition studies for a while, e.g., to show how gaze-indexed frames are beneficial (Li et al., 2015) and how noisy information can be recognized with minimum error (Li et al., 2018). However, using gaze information for VQA studies has not become popular yet.

While VQA datasets have involved video with subtitles (Tapaswi et al., 2016; Lei et al., 2018, 2020), GazeVQA does not depend upon the additional explanation. AGQA (Grunde-McLaughlin et al., 2021) works on spatial-temporal reasoning and NexT-QA (Xiao et al., 2021) deals with temporal-causal reasoning. Assembly101 (Sener et al., 2022) focuses on assembly and disassembly applications, but does not involve collaboration and interaction. GazeVQA works on common grounding for collaborative interactions by using gaze information. It is formulated based on collaborative applications and aims to make predictions with text, image, and video answers.

## 3 Dataset

### 3.1 Video and QA Collection

GazeVQA is a dataset based on simulated scenarios for industrial applications of human-robot collaboration. Figure 2 shows the spatial layout of the data collection. In our use case of industrial assembly and disassembly, the 3D printed version of the Type C Gearbox-AGNEE Shaft Mounted Speed Reducer (SMSR) product was used.

**Task-oriented, instruction-based procedures.** Existing datasets feature multi-step activities following a strictly ordered recipe (Miech et al., 2019; Zhou et al., 2018; Zhukov et al., 2019), scripted (Ragusa et al., 2021); or non-scripted (Sener et al., 2022). GazeVQA consists of instruction-based task-oriented collaborative activities. In particular, the assembly/disassembly was performed by two partners in a collaborative relationship. One partner acts as an instructor, and the other is a novice/subject to be trained. During the experiment, the subjects can ask for clarification and help from the instructor. The subject can be warned and assisted by the instructor in cases of incomplete/incorrect actions.

**Synchronization of exo/egocentric angles.** Fixed features of the scene or objects in the video that do not change over time are called static information, such as the shape of an object or the layout of a room. Such information is often important for understanding the context of a scene or the properties of an object. Information that expresses properties that can change over time is called dynamic information, such as the movement of objects or humans, the changing expressions on a human's face, or the actions performed by the subject of the video. Capturing dynamic information enables the analysis of events or changes in the scene over time. The FVP video provides a subjective view of the action. It can offer an insight into a human's focus of attention. Some tasks are designed for the interaction of multiple objects, such as screwing bolts onto the gearbox. In this example, the gearbox should be put horizontally, and then the nut and bolt should be assembled in the correct matching position. In challenging steps where multiple actions are performed simultaneously, FPV videos cannot provide sufficient data for the assistive model to estimate human intention. Therefore, static and dynamic information might be missed by FPV. We propose that eye-gaze information is the key to understanding the semantic

information from that kind of action in the videos.

VQA studies usually require a large number of QA lists to train a model. Questions are asked to assist the subjects with the necessary information to perform the task. To do so, some questions are prepared by using specific verbs such as "look", and "gaze at". They are used in the dataset to utilize eye-gaze information. In short, questions are prepared to mitigate challenges in temporal action segmentation, action recognition, and object recognition, which can be answered by using gaze information, as well as TPV as an enhanced approach to improving our VQA dataset.

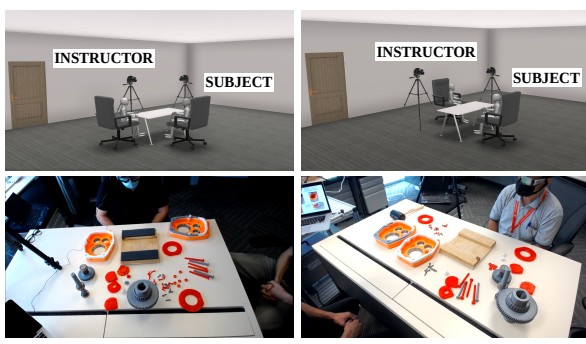

Figure 2: Spatial layout for GazeVQA data collection, and the images of QA tasks with the opening scene.

**Different Perspectives.** None of the 22 participants is knowledgeable about this industrial product. Most experiment sessions were recorded from three different camera angles (one FPV and two TPV angles), except two sessions that were recorded by two camera angles (one FPV and one TPV) due to the malfunctioning of a TPF camera. The use of these different videos has five different goals. First, TPV videos are positioned as shown in Figure 2, providing us with spatial depth. As a future study projection, this depth provided by the angle differences of the cameras can also be used for 3D environment extraction. Second, it is used to spatially locate small components that might be invisible from an angle. Third, the frames taken from FPV videos might not contain the answer to the question asked. This is because the FPV camera has a limited field of view and the subjects constantly move the head from side to side, leading to partial coverage of the scene. Fourth, FPV captures the user's perspective which reflects his/her attention. Finally, with eye gaze in FPV videos, which provides additional information on the user's attention and intention. It helps to model to answer questions such as "Where did the subject

look at?" and "Which object will be assembled after the current object is assembled?"

There are variations in the task completion process where the sequence may deviate from standard ones. For instance, "screw in the oil level indicators" should be completed in one action. In some cases, the subject realizes that as separate sequential tasks. Therefore, the fixed type of QA list cannot be used. However, QA lists could be integrated into the new actions manually. For example, some of the subjects applied the actions "Put the input hub cover" and "Put the small hub cover" at the same time. Thus, the questions are updated for the combination of these steps. Additionally, there are cases where alternative sequences are performed by a subject, e.g., putting the small hub cover before the input hub cover. The detailed visualization is shown in Figures 10 and 11 (in the appendix part).

## 3.2 GazeVQA Task Formulation

For collaborative tasks to be effective, communication (information sharing) is crucial (Tan et al., 2020). The collaborative interaction of instructor and subject has two phases, which are the presentation and acceptance phases according to the common ground theory (CGT) (Clark and Brennan, 1991; Li et al., 2006). CGT requires searches and updates of common ground to maintain effective communication. This continuous process during communication requires mutual agreement, constant attention to feedback, and the adjustment of communication strategies according to the context. The concept is used to analyze the efficiency of communication in different settings, including interpersonal communication and human-computer interaction. In our experiments, the instructions from the instructor (speaker) define the presentation phase, while the listener's understanding of the given instructions and the execution of the instructions defines the acceptance phase.

We formulate a new QA task involving collaborative interaction processes that exist between human-robot/human. The aim of this interaction is to understand the instructions given by the instructor and to implement them faithfully as a sequence of atomic actions. More than 1100 questions are prepared for each participant following guidelines of 13 different reasoning types that are commonly used in VQA studies. Interested readers may refer to Figure 9 in the appendix for a detailed list of reasoning types. The subject can be assisted by

not only textual answers but also video retrieval and/or image answers depending on the types of questions.

Similar to other VQA datasets, we propose a multiple choice (MC) answering format for our textual responses. Since video retrieval answers have been recorded from 3 different video angles (TPVenface, TPVsideward, and FPV - with and without gaze information), 4 different video retrieval responses are provided for each participant (except for two subjects). Additionally, manually created ground truth (GT) image answers with the help of segment anything model (SAM) (Kirillov et al., 2023) are prepared for comparing the gaze location in the frame to generate the image answer.

## 3.3 GazeVQA Statistics and Analysis

The GazeVQA dataset is composed of the assembly process, which includes a maximum of 22 steps, and the disassembly process, which includes a maximum of 19 steps (not symmetrical). The number of questions for the assembly task is 631; and the numbers of textual, image, and video answers are 563, 306, and 479, respectively. The number of questions for the disassembly task is 521, while the numbers of textual, image, and video answers are 474, 266, and 389, respectively.

There is a total of 25,040 questions, which can be broken down into 8,509 "What", 3,017 "Which", 2,798 "Where", 785 "When", 44 "Who", 2,392 "How", 7,407 "Did", 44 "Is" and 44 "Was". The number of unique textual questions is 1,091. The detailed analyses of the QA pairs are shown in the appendix. In total, there are more than 22,000 textual answers, 12,400 labeled images that are used by one participant's video clips, and 2,967 video clips in the dataset. Additionally, 18,700 video answers are expected, according to the questions. Statistics on frequent words and questions are provided in Figure 5 and 6 in the appendix.

## 4 Model

To tackle the challenges presented in GazeVQA, especially related to the gaze-augmented multimodal information, we propose a new AssistGaze model as shown in Figure 3.

The first critical challenge relates to the encoding of multimodal inputs, especially video input. Video encoding serves the purpose of obtaining object information in video embedding (Wang et al., 2022b,a). Text encoding involves question encod-

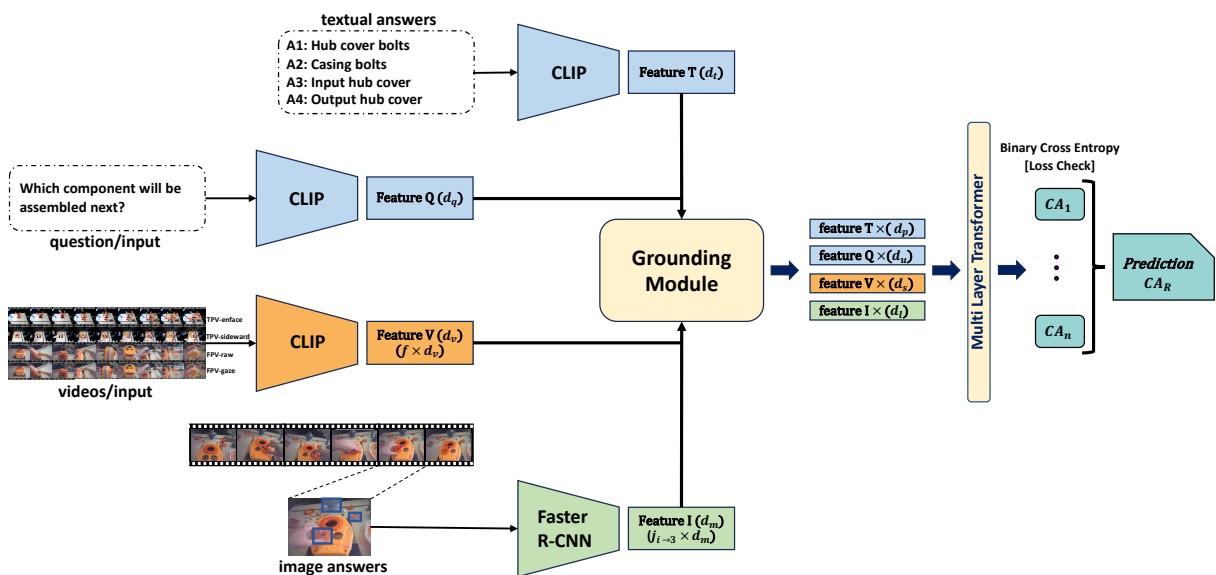

Figure 3: AssistGaze Model. Firstly, the features of questions and videos are taken as inputs; additionally, the features of textual answers and labeled images are extracted. Then the encoder structure is where the feature representations are obtained by input encoders. Finally, it performs answer prediction by contrastive loss check.

ing and text answer encoding. CLIP is used for the text embedding. This ensures that the video features and the text features are projected onto the same dimensional space. Each question has a shape of $1 \times d_q$ and the text answer will have a shape of $1 \times d_t$, where $d_q$, and $d_t$ are the dimensions of features. The model needs to select the correct text tensors from the $m$ choices provided.

For the video encoder, CLIP (Radford et al., 2021) is used to extract the features. The feature extractions from the video clip consist of a sequence of five distinct phases. We sparsely sampled 8 frames from each action step and encoded each frame with CLIP. The feature of each frame has a shape of $1 \times d_v$ where $d_v$ is the encoding dimension. Each video answer consists of four videos of which there are two FPV videos and two TPV videos. One of the FPV videos is the raw version without gaze information and the other is the gaze-augmented version, where the gaze point is represented by a blue dot. While four phases represent the feature extraction of four different types of videos, the last phase represents the concatenation of these four views. The model needs to select the correct step from the 5 steps in the video.

The Faster R-CNN (Ren et al., 2015) is implemented for image encoding. Since the task requires the model to select the correct bounding box out of the "**n**" choices. To get the features inside the bounding box, the Faster R-CNN is suitable due to

the Region of Interest (ROI). It can crop the image within the bounding box and convert it into features with the shape of $1 \times d_m$.

After all features are extracted, **question features** $q \in R^{1 \times d_q}$, **video features** $v \in R^{5 \times d_v^c}$, **video answer features** $a \in R^{1 \times d_v^c}$ with **each image answer features** $1 \times d_m$ and **each text answer feature** $t \in R^{1 \times d_t}$ are concatenated. These features are called one-question-answer tuples. The concatenated feature for each question-answer tuple $m \times n \times l$ is with a shape of $8 \times d_f$, where $d_f$ is the feature dimension.

Each tuple will be passed through a 3-layer transformer encoder and we only use the class token of the output tensor from the transformer encoder. It has a shape of $1 \times d_f$ and then it will be passed through a fully connected layer with the shape of $d \times 1$. This will give us a prediction $p$ of whether this pair is correct or incorrect.

A notable challenge in VQA is short-cut learning, which has been addressed by various strategies, such as acting on training data, different learning strategies, and architectural priors (Dancette, 2023). A more balanced approach is suggested in (Goyal et al., 2017), where the VQAv2 dataset is designed to make it difficult to answer a question using only the image. (Johnson et al., 2017), and (Hudson and Manning, 2019) have tried to reduce conditional biases with the rejection sampling approach. We tackle this issue by designing

| Methods | Answering Type | mAP |
|---|---|---|
| Random Guess | Text | 0.0711 |
| VIOLET (Fu et al., 2021) | Text | 0.2549 |
| (Ours) 8 Frames of CLIP Features | Text | 0.2717 |
| (Ours) 16 Frames of CLIP Features | Text + Image + Video | 0.4919 |
| (Ours) FPV Video Add TPV Video | Text + Image + Video | 0.5040 |
| **(Ours) 8 Frames of CLIP Features (standard)** | Text + Image + Video | **0.6719** |

Table 2: The results for baseline methods which are prepared in different frames with CLIP (Devlin et al., 2019) features and FPV video additions to the TPV videos. The benchmark has been done with the VIOLET (Fu et al., 2021) model by considering only the textual answer type. Not only does our novel multiple-answering approach, but our single-type answering approach also performs better against the SOTA textual answer models.

GazeVQA and AssistGaze with multi-modality answers (text, image, or video). We also prepared a number of counterfactual QA pairs to reduce the possibility of shortcut learning, as inspired by (Goyal et al., 2017; Ramakrishnan et al., 2018).

## 5 Experiments

**Data Splits.** GazeVQA consists of a total of 3,008,400 question-answer tuples. These tuples are randomly split into the training set and validation set with a ratio of $8:2$.

**Evaluation Metrics.** For each question we have $m \times n \times l$ question-answer tuples. Among these tuples, there is only one tuple that can be labeled as correct. Specifically, an answer is considered correct only if all three types of answers (text, image, video) are correct; otherwise, it is labeled as wrong. Therefore, only if all answers are answered correctly, then the prediction will be labeled as correct with a ratio of positive samples to negative samples being $1 : m \times n \times l - 1$. We use mean Average Precision (mAP) as the primary evaluation metrics (Oksuz et al., 2021).

$$Precision = \frac{TruePstv.}{TruePstv. + FalsePstv.} \quad (1)$$

**Implementation Details.** We use PyTorch (Paszke et al., 2019) to perform the experiments. The AdamW (Loshchilov and Hutter, 2019) optimizer is used with learning rate of $6 \times 10^{-7}$. The batch size is set to 1024.

### 5.1 Ablation Studies and Results

We first show the performance comparison of a few models. As shown in Table 2, random guess has an mAP of 0.0711. Our standard baseline uses 8 frames of one video to be the representation of the

video, which is suggested by ClipBERT (Lei et al., 2021). As shown in Table 2, the mAP result which is 0.6719 reflects that 8 frames perform well in our task and more frames, 16, even undermine the performance of the model 0.4919. In the standard model, the input modality includes two TPV videos and two FPV videos, which achieves an mAP of 0.6719. To evaluate the contribution of different features, we test a few ablated feature combinations. For example, we use the FPV video with the TPV video with a trainable hyper-parameter $\alpha$ and get a finalized video features $\alpha \times 2 \times d_v^f + (1-\alpha) \times 2 \times d_v^s$, the mAP decreases to 0.5040. As shown in Table 2, with only part of the features, the performance is lower than the standard model.

| Video Inputs | | | Metrics |
|---|---|---|---|
| FPV | TPV | EyeGaze | mAP |
| ✓ | ✓ | ✗ | 0.5004 |
| ✗ | ✓ | ✓ | 0.5061 |
| ✓ | ✗ | ✓ | 0.4982 |
| ✓ | ✓ | ✓ | 0.6719 |

Table 3: Ablation studies and results for the usage of different multi-view inputs (FPV, TPV, and Eye-Gaze).

The above results show the challenging nature of the GazeVQA problem. It requires varying modalities of answers while existing SOTA models require only single-modality answers. As shown in Tables 3 and 5, the performance is lower when only one or two kinds of answer types are involved (Note that the question and video features are preserved). The lowest mAP (=0.2274) is obtained when only the video feature is used. However, the performance of our model is more suitable and better when multiple answer types are needed.

To evaluate the effect of different encoders, we use MViTv2 (Li et al., 2022) extract video fea-

| Epoch | Learning Rate:1.2e-5 mAP | Learning Rate:8e-4 mAP | LR:8e-4, Unfrozen Layer of BERT mAP |
|---|---|---|---|
| 1 | 0.0939 | 0.1047 | 0.2318 |
| 2 | 0.0977 | 0.1042 | 0.2542 |
| 3 | 0.1006 | 0.1039 | 0.2506 |
| 4 | - | 0.1056 | **0.2549** |

Table 4: The ablation studies that are conducted by using the VIOLET Model (Fu et al., 2021). It shows the Fine-tuning of the VIOLET model's fully connected layer with different learning rates. The last column represents the fine-tuned linear layer of the VIOLET (Fu et al., 2021) by unfreezing the last layer of the BERT (Devlin et al., 2018). LR: Learning Rate.

| Answer Types | | | Metrics |
|---|---|---|---|
| Video | Text | Image | mAP |
| ✗ | ✗ | ✓ | 0.3321 |
| ✗ | ✓ | ✓ | 0.4482 |
| ✗ | ✓ | ✗ | 0.2717 |
| ✓ | ✓ | ✗ | 0.2987 |
| ✓ | ✗ | ✗ | 0.2274 |
| ✓ | ✗ | ✓ | 0.3523 |
| ✓ | ✓ | ✓ | 0.6719 |

Table 5: The results of the usage of different answer types which are video, textual, and image.

tures and BERT (Devlin et al., 2018) to extract text features (Devlin et al., 2019), as compared to the CLIP-based model. Each step in a video is encoded with MViTv2 and we take the class token of the features as the representation of that step. Since MViT is not co-trained with BERT in the same manner as in CLIP, the model faces challenges in determining the relationships among multimodal features. The performance analysis and comparison between MViT, BERT, and CLIP are displayed in Table 6.

| Encoders | | | Metrics |
|---|---|---|---|
| Video | Text | Object | mAP |
| MViTv2 | BERT | Faster R-CNN | 0.3993 |
| CLIP | CLIP | Faster R-CNN | 0.6719 |

Table 6: Ablation studies on feature encoders to determine the use case differences of CLIP and MViTv2 for textual and video features, while Faster R-CNN is used for object features.

Furthermore, the encoder plays a critical role in the model. As shown in Table 7, the mAP result is 0.4557 with 2 layers of the encoder. Additionally, the mAP value is 0.5289 with 4 layers of the

encoder. Finally, when the encoder layer is set to 3, the model performance achieves the best result, with an mAP of 0.6719.

| Metrics | # of Encoder Layers | | |
|---|---|---|---|
| | 2 | 3 | 4 |
| mAP | 0.4557 | 0.6719 | 0.5289 |

Table 7: Ablation studies for the usage of different numbers of encoder layers to get accurate results.

Next, we test the effectiveness of the recent SOTA model VIOLET (Fu et al., 2021) on our dataset. VIOLET supports only textual answer type and we compare it with our method on textual answer results. VIOLET uses a Video Swin Transformer (Liu et al., 2022) to encode the images and BERT (Devlin et al., 2018) to encode the text. We fine-tuned the classification head of the VIOLET model while keeping the parameters of the Swin Transformer and BERT frozen. However, the mAP plateaued at 0.10 (as shown in Table 4), indicating the limited performance of VIOLET on our dataset when only fine-tuning the classification head.

To better align the features with our task, we unfroze the last layer of the BERT encoder. After four epochs of training, the mAP of the VIOLET model converged to a value of 0.2549 which is slightly lower than our baseline result. Comprehensive ablation results are shown in Table 4.

Finally, the benchmarking results of textual answers with the VIOLET model are given in Table 2. AssistGaze employs the CLIP to encode both image and video features, while VIOLET uses the Swin Transformers for video encoding and the BERT for text encoding. CLIP efficiently learns visual concepts from natural language supervision. Consequently, using the CLIP as an encoder may result in better performance in visual question-answering tasks.

# 6 Conclusions

In this paper, with a new dataset GazeVQA, we propose a novel approach to task-oriented collaborative QA towards real-world applications like human-human/robot collaboration. The GazeVQA depicts the natural QA collaboration interaction between an instructor and a subject during guided assembly and disassembly tasks. Moreover, we propose a new method with a multiple-choice answer format for all types of questions designed to perpetuate the common grounding theory for collaboration tasks. Unlike the legacy VQA studies, we address a challenging task in the industrial settings on collaborative task execution. The proposed AssistGaze is a new baseline for evaluating the VQA model's ability to address this challenging task.

In future studies, further developments are required to improve the collaboration of computer vision with semantic understanding to inquire into the collaboration of AssistGaze and GazeVQA on real-world collaborative HRI applications. More efforts are needed to address high-level challenges such as temporal action segmentation, action recognition, object recognition, and spatial understanding tasks. We hope that the proposed novel gaze-enhanced VQA dataset "GazeVQA" and the new assistant model "AssistGaze" will enable the community to move forward with task-oriented VQA.

## Limitations

The current work has the following limitations: (1) The GazeVQA dataset was designed with a new task-oriented perspective, which is a novel part of the dataset. However, the existing models need to be updated in their architectures to utilize the advantages of GazeVQA. (2) The number of tasks is limited, and this limitation could indicate a challenge for the other collaborative applications. (3) Like many gaze-related researches that use mobile eye-tracker, gaze information in the current dataset is still noisy. Novel algorithms to de-noise the gaze data are needed to enhance the consistency and validity of the dataset.

## Acknowledgement

This research is supported by the Agency for Science, Technology and Research (A*STAR) under its AME Programmatic Funding Scheme (Project A18A2b0046). The NUS team is supported by the National Research Foundation, Singapore under its NRFF Award NRF-NRFF13-2021-0008, and Mike Zheng Shou's Start-Up Grant from NUS.

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

# A  Appendix

GazeVQA has been designed to offer a fundamental and generalizable approach to human-human and human-robot interaction, especially for industrial collaborative task studies. AssistGaze proposes an assistive model that can support people in task-collaborative studies. The purpose of design protocols is task completion by using eye-gaze information. It is predicted that studies such as human intention estimation and next action estimation with over 3,000,000 QA pairs and gaze location information will also benefit robotic applications. Both contributions of this paper are essential for task-oriented human-robot interaction with the question-answering project.

## A.1  GazeVQA Architecture

GazeVQA consists of assembly-disassembly videos obtained from 22 users, questions prepared separately for each action performed by each user, and labeled images from FPVs which are coordinated with eye-gaze information.

**Video Clips.** Each user performed an industrial task by following the instructions given by an instructor. The users' videos were cropped as 5 actions were performed consecutively. This is designed to reduce the processing cost of long videos to more optimum values. Moreover, videos with common steps increase the variety of uses in response to the questions asked.

**QA Pairs.** We have also published the raw versions of our questions. This will facilitate a better understanding of industrial QA pairs and the development of textual inputs.

**Images.** Each of the labeled images was prepared manually using the Segment Anything Model (Kirillov et al., 2023). When the eye-gaze data is on the object related to the answer to the question, the object or that answer is labeled as GT. It also contains two different incorrect labels.

- Figure 4 shows the GazeVQA Architecture.

## A.2  Words and Questions

One of the motivations for preparing QA pairs is to focus on the use of different verbs, words, and question types. Considering the small number of industrial datasets, the textual QA pairs in GazeVQA are a useful supplement to existing studies.

- Figures 5 and 6 show the diagram of most frequent words and questions.

## A.3  VQA Datasets

One of the other motivations is the lack of datasets prepared for task-oriented collaborative scenarios. GazeVQA leverages industrial collaborations by closing this gap and taking advantage of different targets, video sources, and purposes.

- Tables 8 and 9 display the comparison of the GazeVQA dataset and existing VQA datasets.

- Figure 9 displays the comparison of the GazeVQA dataset with existing VQA datasets in terms of reasoning types.

## A.4  Assembly and Disassembly

Assembly and disassembly are crucial tasks for industrial applications. This leverages another motivation to create a new dataset for real-world task-oriented collaborations.

- Figures 7 and 8 show the assembly and disassembly actions with synchronized multi-view video clips.

- Figure 10 and 11 show the assembly and disassembly steps and protocols.

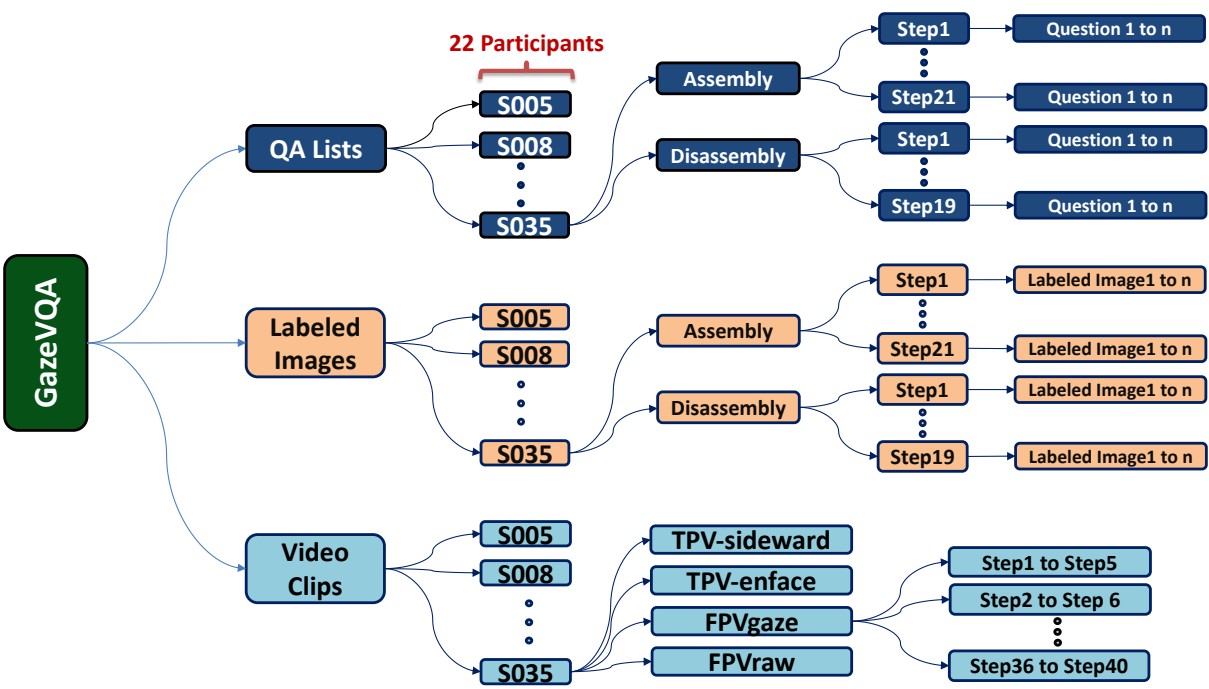

Figure 4: GazeVQA Architecture. It consists of three different parts. QA Lists, Labeled images, and Video Clips. There are 22 participants, and each participant has their own sub-structures that are divided into assembly and disassembly parts for QA lists and Labeled images. additionally, they have 4 different video clips (TPV-sideward, TPV-enface, FPVgaze, and FPVraw) which explain why they were collected separately in the Dataset section of the paper. Each step has its own QA lists, labeled images, and video clips.

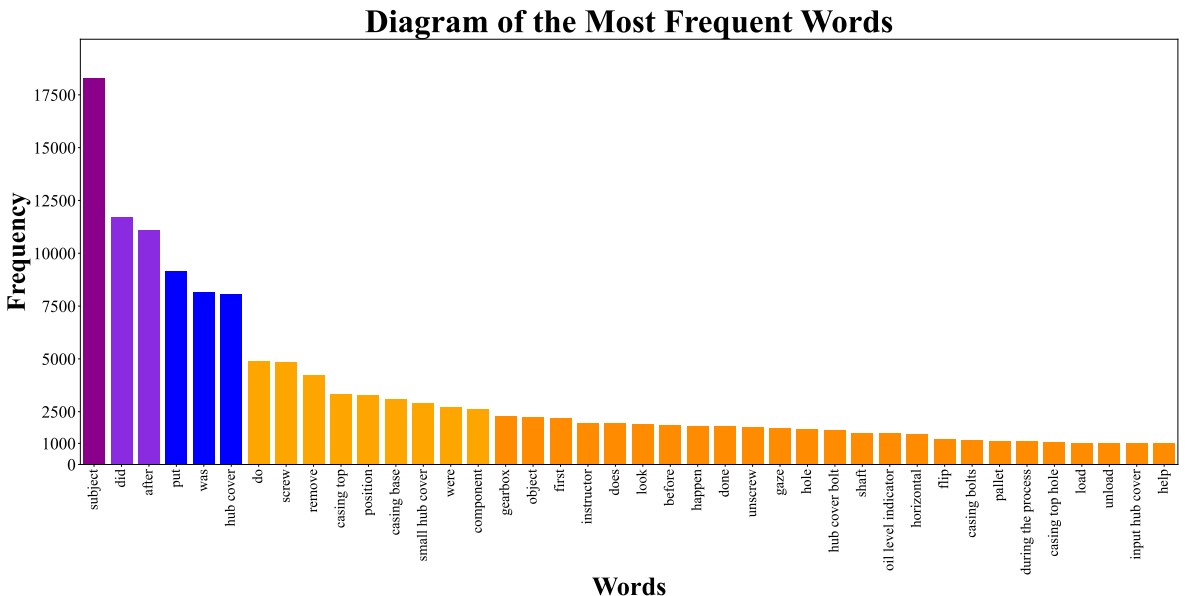

Figure 5: Diagram of the Most Frequent Words of the GazeVQA. "Subject" is the word that is most frequently used in the dataset as a subject, and then "did" is used to check whether the actions/tasks are completed or not. If the diagrams or the datasets are checked more than 20 action verbs are used, and their frequency is moderately higher than the other words used. The reason is that the usage of more verbs could be helpful to increase the possibility of the usage of the GazeVQA QA list on the other industrial projects' applications. Consequently, it could increase the generalizability of the GazeVQA.

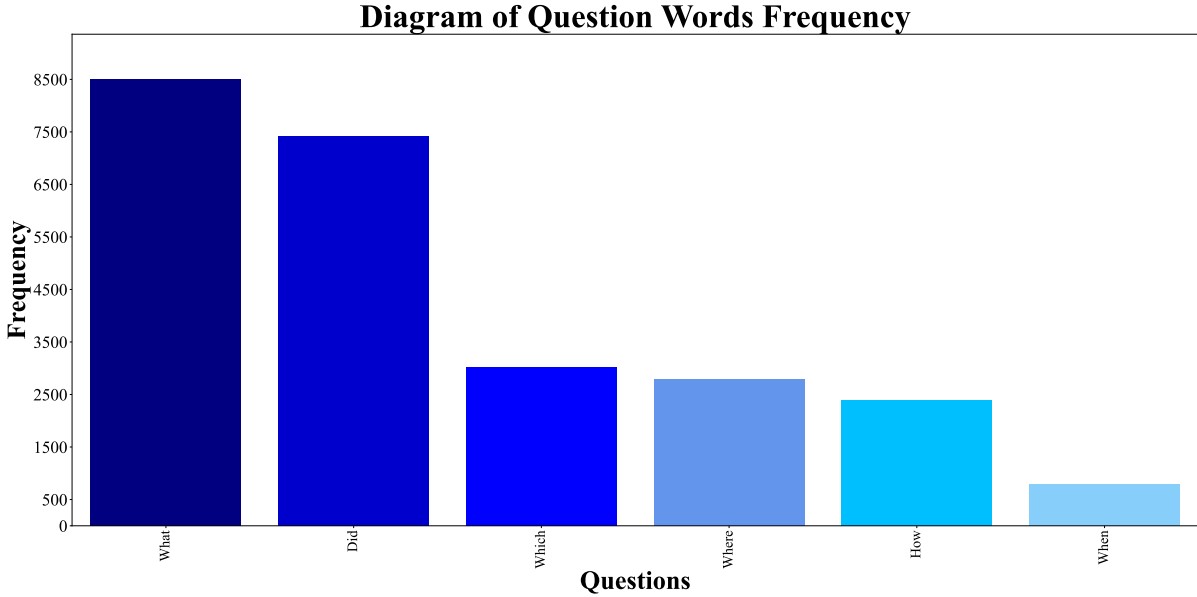

Figure 6: Diagram of the Question Frequency of the GazeVQA. "What" and "Did" are the most frequent questions that are used to prove the application realization, object, and action recognition. On the other hand, "Which" and "Where" questions are helpful to object localization to increase the answers' accuracy of the spatial questions. The importance of the "How" question is to show the completion of the tasks.

| Datasets | Video Source | # of V. | # of QA | Annota. |
|---|---|---|---|---|
| MSVD-QA (Xu et al., 2017) | Web Videos | 2K | 50K | Auto. |
| TVQA+ (Lei et al., 2020) | TV Programs | 22K | 153K | Manual |
| TutorialVQA (Colas et al., 2020) | Instructional Videos | 408 | 6.2K | Manual |
| ActivityNet-QA (Yu et al., 2019) | Web Videos | 5.8K | 58K | Manual |
| CLEVRER (Yi et al., 2020) | Synthetic | 10K | 305K | Auto. |
| AGQA (Grunde-McLaughlin et al., 2021) | Indoor Activity Videos | 9.6K | 192M | Auto. |
| NExT-QA (Xiao et al., 2021) | Web Videos | 5.4K | 52K | Manual |
| HowToVQA69M (Miech et al., 2019) | Instructional Web | 10K | 10K | Auto. |
| **GazeVQA (Ours)** | **Task Collaboration** | 2,967 | 3M | Manual |

Table 8: Comparison of GazeVQA and existing VQA datasets by evaluating the number of videos, QA pairs, and annotation types.

| Datasets | Target | Text A. | Im. A. | V. A. |
|---|---|---|---|---|
| MSVD-QA (Xu et al., 2017) | Description | ✓ | ✗ | ✗ |
| TVQA+ (Lei et al., 2020) | Subtitle inclusion | ✓ | ✓ | ✓ |
| TutorialVQA (Colas et al., 2020) | Instructional Assistance | ✗ | ✗ | ✓ |
| ActivityNet-QA (Yu et al., 2019) | Description | ✓ | ✗ | ✗ |
| CLEVRER (Yi et al., 2020) | Causal Reasoning | ✓ | ✗ | ✗ |
| AGQA (Grunde-McLaughlin et al., 2021) | Compositional Reasoning | ✓ | ✗ | ✗ |
| NExT-QA (Xiao et al., 2021) | Causal-Temporal Relation | ✓ | ✗ | ✗ |
| HowToVQA69M (Miech et al., 2019) | Generalization | ✓ | ✗ | ✗ |
| **GazeVQA (Ours)** | **Common grounding for collaboration by gaze** | ✓ | ✓ | ✓ |

Table 9: Comparison of existing VQA datasets and GazeVQA which targets to obtain all answer formats with Multiple Choices (MC) answer types by using novel gaze information.

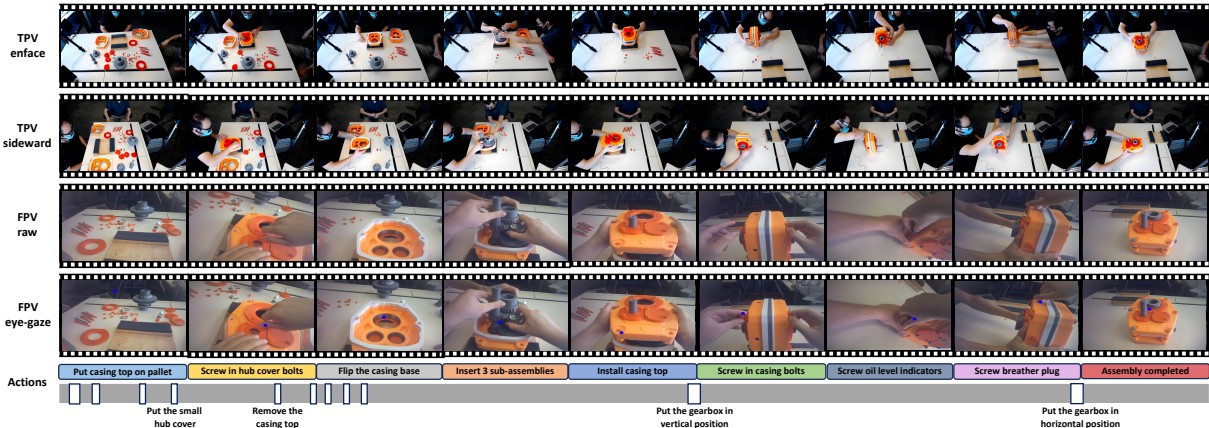

Figure 7: GazeVQA assembly action includes synchronized multi-view video clips of assembling the gearbox.

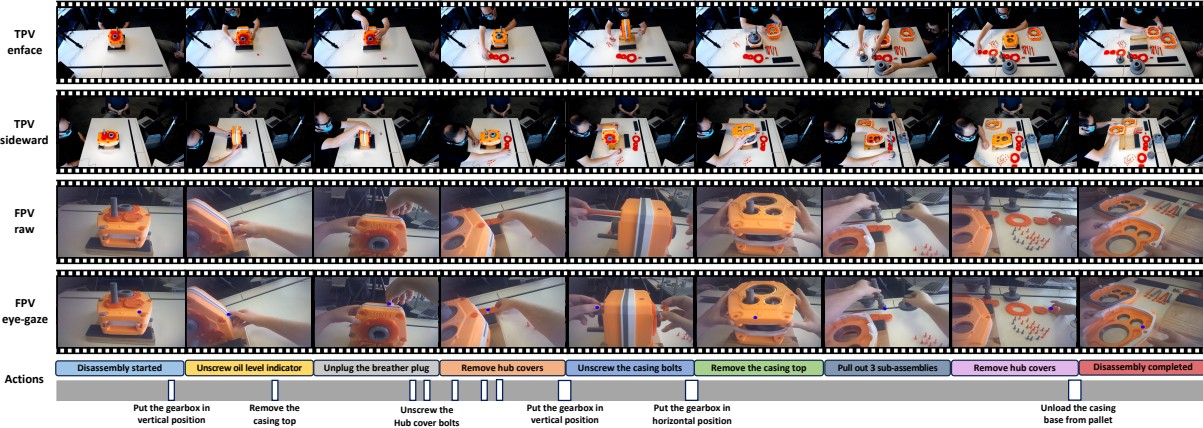

Figure 8: GazeVQA disassembly action includes synchronized multi-view video clips of disassembling the gearbox.

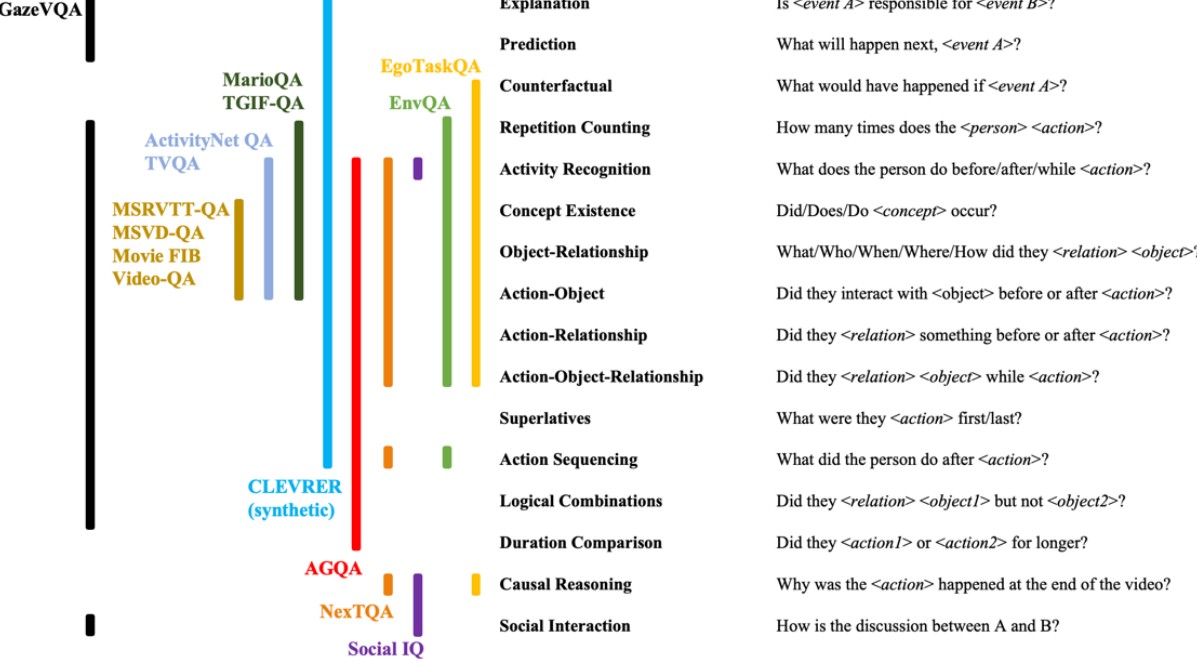

Figure 9: GazeVQA focuses on visual, semantic, spatial, location, temporal, and action-object relationships, as well as social interaction information extracted from questions that are asked from 13 different reasoning types. Figure 9 is prepared to compare the current dataset by benefiting from AGQA (Grunde-McLaughlin et al., 2021).

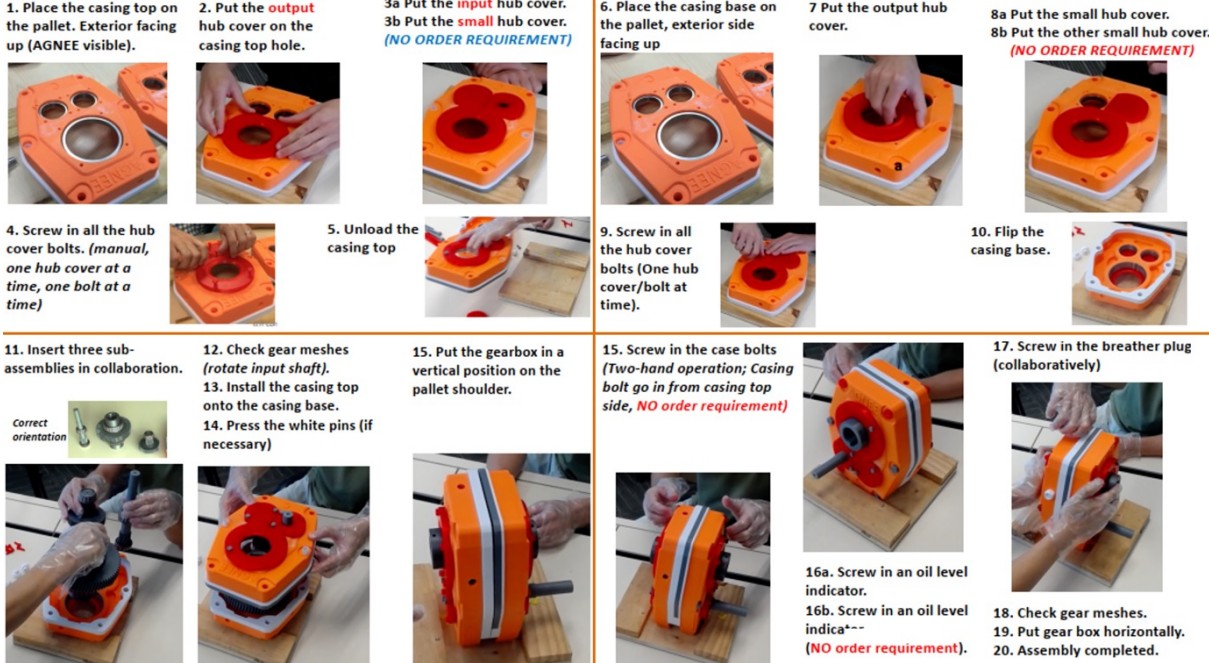

Figure 10: Assembly Steps and Protocols: The assembly task has 20 steps. However, we deleted the "pressing the white pins" step, which is 14, because it was not a common task among the subjects. Some tasks are separated and realized by the subject in two different steps. For instance, there are two oil level indicators, so screwing two oil level indicators is shown as one step in the figure, but some subjects have realized this in two steps. Thus, we have a maximum of 22 steps in the assembly task.

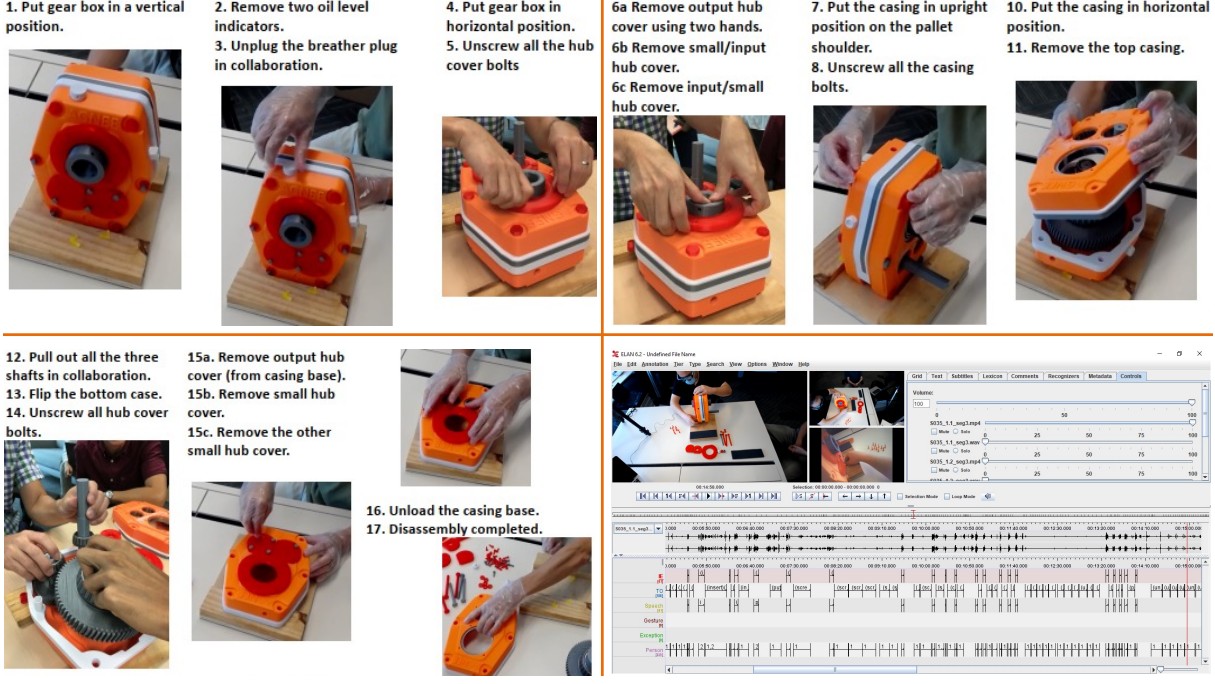

Figure 11: Disassembly Steps and Protocols: The disassembly task has 17 steps. However, some tasks are separated and realized by the subject in two different steps. For instance, there are two oil level indicators, so removing two oil level indicators is shown as one step in the figure, but some subjects have realized this in two steps. Thus, we have a maximum of 19 steps in the disassembly task. The picture on the right-bottom is the annotation example to show how to annotate the steps as atomic actions.