# OpenReview forum: "GazeVQA: A Video Question Answering Dataset for Multiview Eye-Gaze Task-Oriented Collaborations"
_EMNLP/2023/Conference — EMNLP 2023 Main_

### Official Review · Reviewer_doaa · 2023-07-31

**Typos Grammar Style And Presentation Improvements:** 1. The are many ungrammatical sentenc…
**Soundness:** 3

**Excitement:**

3: Ambivalent: It has merits (e.g., it reports state-of-the-art results, the idea is nice), but there are key weaknesses (e.g., it describes incremental work), and it can significantly benefit from another round of revision. However, I won't object to accepting it if my co-reviewers champion it.

**Paper Topic And Main Contributions:**

This work defines a new VQA task with video input from multiple viewing angles and constructs a video dataset from new recorded videos tailored to the task. The tasks focus on assembly and disassembly of a product, with fpv and tpv videos recorded. The authors also implement a naive baseline fusing together video data and question/choice features in a multilayer transformer.

**Questions For The Authors:**

A. Can you further explain what is the "Grounding Module in Figure 3?

**Reasons To Accept:**

1. The new dataset is large in size and video/image quality seems high. The introduction of this new resource can help promote research in surrounding fields.

**Reasons To Reject:**

1. The generalizability of this work is questionable as it only contains videos of assembly and disassembly tasks. Furthermore, all the examples in the paper seem to be conducting assembly and disassembly on the same object? and in the same environment? If this is the case for the whole dataset then generalizability will be a very serious issue.
2. Throughout the paper, there are many incoherent sentences such as detailed in the grammar/typo section below, which can be quite confusing for readers. While this is less of a problem if just in paper writing, the question in the example in Fig.1 "What was put by the subject after the output hub cover was put?" is also not grammatical. This problem could potentially affect the benchmark evaluation if frequently present in the dataset (No code nor data is provided with the submission so its not possible to verify). Thus, the paper and dataset will likely benefit from another round of revision/validation.
3. From the broad explanation in the introduction (L72-82) alone, it is unclear how Gaze data actually benefits the VQA task in real world scenarios. It would help if the authors can back up this claim with concrete examples from the dataset.

**Reproducibility:**

3: Could reproduce the results with some difficulty. The settings of parameters are underspecified or subjectively determined; the training/evaluation data are not widely available.

**Reviewer Confidence:**

4: Quite sure. I tried to check the important points carefully. It's unlikely, though conceivable, that I missed something that should affect my ratings.

---

> ### Author Rebuttal · Authors · 2023-08-29
>
> ## Rebuttal for Reviewer doaa
>
> I would like to clarify the concerns of the reviewer in three aspects that are generalizability of the dataset, the application of the gaze data in real-world scenarios, and grammar improvement.
>
> ---
>
> ### **W1: Generalizability of the dataset**
>
> * __Assembly and disassembly tasks are processes within industrial tasks, encompassing a wide array of significant scenarios__.
>     * __They are foundational scenarios frequently encountered in various industrial applications__.
>     * While the experiment process is similar across users, many variations which are **"the layout of the components, users’ diverse styles to complete the steps, and interaction between instructor and user,"** exist in GazeVQA. Therefore, we focus on striking a good balance of procedure consistency and variations.
>     * __These highlight the reason why assembly-disassembly serves as fundamentals for a wide range of scenarios__.
>
> * __Dataset comparisons__
>     * EPIC-KITCHEN [1], and Breakfast Actions Dataset [2] are niche studies focused on daily kitchen actions.
>     * MECCANO [3] and Assembly101 [4] are also studies that focus on the same environment and objects.
>     * Unlike MECCANO [3] and Assembly101 [4], __GazeVQA uses a one-to-one scale 3D version of real-world products__.
>     * __GazeVQA has a high potential to be used in real-world applications__ with more than 125 hours of video clips, more than 40 objects, and steps.
>     * The generalizability concern of the reviewer could be mitigated by the examples that were given to the reviewer Ab7b
>         * Video clips are collected in 3 different video angles (125 hours). More than 40 components and action steps have existed in GazeVQA.
>
> For evaluation in terms of task-oriented collaborative studies, GazeVQA can be generalizable for numerous collaborative studies such as **human-robot interaction (HRI) studies, human co-worker task assistance, task-assisted video question answering (VQA), and human intention prediction**.
>
> ---
>
> ### **W2: Applicability of the dataset**
>
> * The recent SOTA model “VIOLET” [9] (100+ citations to date) which has only textual answer type, is used for benchmarking.
>     * VIOLET uses a Video Swin Transformer [10] to encode the images and BERT [11] to encode the text.
>     * Due to the limited time period for rebuttal, the VIOLET [9] model with its pertained weight with 1 epoch was fine-tuned.
>     * __We intend to provide updated results during the designated discussion period__, once model convergence through the fine-tuning process. __Results are shown in the Table below__.
>
> | Methods        |Answering Type | mAP                    |
> | -------------  | ------------- |-------------           |
> | Random Guess   |Text           | 0.07117 (from paper)   |
> | VIOLET [5]     |Text           | 0.09396 (to be updated)|
> | Ours           | Text          | 0.27173 (from paper)   |
> | Ours           | Text + Video + Image | 0.67199 (from paper) |
>
> * __We only fine-tuned the classification head of the VIOLET [9] model, while keeping the parameters of Swin Transformer [10] and BERT [11] frozen due to the time and computation resources limit__.  Hence the VIOLET model is underperforming compared to our results.
>
>
> * __Review for applicability of the gaze information__
>     * Head/hand motion and hand location/pose [5],
>     * Sensors in virtual environments [6]
>     * AR/VR studies [7, 8]
>
> * Gaze information can help to predict which object the user will interact with in the next step by recognizing relevant action (as shown in Fig. 1).
>
> * Gaze information can help multi-modal alignment with high precision for ensuring to processing of the video in real time.
>     * __As shown in Fig. 1, gaze information consists of valuable information to answer questions__ such as “Where does the user gaze at”, “looking at”, “Which component will be assembled next?”, or "How to assemble <next object>?".
>     * Gaze information __ensures to location of the objects and components with the help of detailed labeling as shown in our study.__ Thus, it obtains better performance for object recognition or action recognition tasks.
>
> * For the grounding module, it could be mentioned that the Image encoded by Fast RCNN has a different length compared to the clip features within the modules. First, we need to convert them into the same dimensions, like a Conv1D layer to make them have the same length. Then we get the all-feature extractions to put in a multilayer transformer, before the loss check.
>
> ---
>
> ### **W3: Grammar style and improvements**
>
> * __Ungrammatical Sentences__
>     * The paper has been read many times and has been checked by using multiple grammar-checking programs. **Mistakes were corrected**, such as:
>         * The incorrect sentence was “The first-person view provides a subjective view of the action and can insights into the human’s focus of attention”
>         * It was corrected by adding the word “offer” before the word “insights”.
>
> * __Dataset Question Styles__
>     * We analyzed a meticulous examination of various question styles and addressed their implications in our dataset. __There are no grammatically incorrect questions but inverted questions prepared to challenge the model__.
>     * The experiments have been done multiple times with different versions of QA pairs. According to our results and efforts for fair benchmarking, we have not encountered a situation that potentially affects the benchmarking.
>
> * __Explicit Captions for Tables__
>     * Finally, I would like to mention that the captions of the tables have been changed as follows:
>         * Table 1: Comparison of the GazeVQA and other action datasets in the context of total hours of video clips, number of videos, average length of video clips (in minutes), number of verbs, actions, and objects.
>         * Table 2: The results for baseline methods which are prepared in different frames with CLIP features and FPV video additions to the TPV videos.
>         * Table 3: Ablation studies on feature encoders to determine the use case differences of CLIP and MViTv2 for Textual and Video features, while Faster R-CNN is used for object features.
>         * Table 4: Ablation studies for usage of different multi-view inputs
>         * Table 5: Ablation studies for the usage of different numbers of encoder layers to get accurate results.
>         * Table 6: The results of the usage of different answer types which are video, textual, and image, individually and collaboratively.
> ___
>
> ### **REFERENCES**
>
> 1. Damen, D., Doughty, H., Farinella, G. M., Furnari, A., Kazakos, E., Ma, J., ... & Wray, M. (2022). Rescaling egocentric vision: Collection, pipeline and challenges for epic-kitchens-100. International Journal of Computer Vision, 1-23.
> 2. Kuehne, H., Arslan, A., & Serre, T. (2014). The language of actions: Recovering the syntax and semantics of goal-directed human activities. In Proceedings of the IEEE conference on computer vision and pattern recognition (pp. 780-787).
> 3. Ragusa, F., Furnari, A., Livatino, S., & Farinella, G. M. (2021). The meccano dataset: Understanding human-object interactions from egocentric videos in an industrial-like domain. In Proceedings of the IEEE/CVF Winter Conference on Applications of Computer Vision (pp. 1569-1578).
> 4. Sener, F., Chatterjee, D., Shelepov, D., He, K., Singhania, D., Wang, R., & Yao, A. (2022). Assembly101: A large-scale multi-view video dataset for understanding procedural activities. In Proceedings of the IEEE/CVF Conference on Computer Vision and Pattern Recognition (pp. 21096-21106).
> 5. Li, Y., Fathi, A., & Rehg, J. M. (2013). Learning to predict gaze in egocentric video. In Proceedings of the IEEE international conference on computer vision (pp. 3216-3223).
> 6. Emery, K. J., Zannoli, M., Xiao, L., Warren, J., & Talathi, S. S. (2021, March). Estimating Gaze From Head and Hand Pose and Scene Images for Open-Ended Exploration in VR Environments. In 2021 IEEE Conference on Virtual Reality and 3D User Interfaces Abstracts and Workshops (VRW) (pp. 554-555). IEEE.
> 7. Rivu, R., Abdrabou, Y., Pfeuffer, K., Esteves, A., Meitner, S., & Alt, F. (2020, June). Stare: gaze-assisted face-to-face communication in augmented reality. In ACM Symposium on Eye Tracking Research and Applications (pp. 1-5).
> 8. Pathmanathan, N., Becher, M., Rodrigues, N., Reina, G., Ertl, T., Weiskopf, D., & Sedlmair, M. (2020, June). Eye vs. head: Comparing gaze methods for interaction in augmented reality. In ACM Symposium on Eye Tracking Research and Applications (pp. 1-5).
> 9. Fu, T. J., Li, L., Gan, Z., Lin, K., Wang, W. Y., Wang, L., & Liu, Z. (2021). Violet: End-to-end video-language transformers with masked visual-token modeling. arXiv preprint arXiv:2111.12681.
> 10. Liu, Z., Ning, J., Cao, Y., Wei, Y., Zhang, Z., Lin, S., & Hu, H. (2022). Video swin transformer. In Proceedings of the IEEE/CVF conference on computer vision and pattern recognition (pp. 3202-3211).
> 11. Devlin, J., Chang, M. W., Lee, K., & Toutanova, K. (2018). Bert: Pre-training of deep bidirectional transformers for language understanding. arXiv preprint arXiv:1810.04805.

---

### Official Review · Reviewer_3C3r · 2023-08-01

**Typos Grammar Style And Presentation Improvements:** This paper is well organized and writ…
**Soundness:** 3

**Excitement:**

4: Strong: This paper deepens the understanding of some phenomenon or lowers the barriers to an existing research direction.

**Paper Topic And Main Contributions:**

The paper's central focus is the incorporation of exocentric and egocentric videos in Video Question Answering (VQA) studies, a cutting-edge approach in robot-human interaction and collaboration studies. The paper's unique contribution lies in the creation of a novel task-oriented VQA dataset, referred to as GazeVQA, which incorporates gaze information during collaborative task processes to understand human intentions.

**Questions For The Authors:**

The dataset is compiled from 22 participants. This sample size might not be adequately representative to capture the diversity and nuances of human gaze and intent in collaborative tasks, leading to potential biases.

The emphasis on gaze data might make the VQA approach presented overly reliant on this particular modality. Real-world applications may encounter scenarios where gaze data isn't as indicative of intent, or where such data is unavailable or unreliable.

How does the utilization of gaze information in VQA differ from its use in action recognition studies, given that it has been employed in the latter for some time?

In Table2, why 8 Frames of CLIP Features achieves better performance than 16 Frames of CLIP Features?

**Reasons To Accept:**

The inclusion of both exocentric and egocentric videos with gaze information provides a unique perspective on human intentions during task processes.

The introduction of the AssistGaze model demonstrates a practical application of the GazeVQA dataset.

The extensive experiments performed showcase the efficacy of AssistGaze and the intricacies of GazeVQA.

**Reasons To Reject:**

While the GazeVQA dataset's innovation is commendable, it addresses a very specific niche within the broader NLP community. The focus on gaze data and its application to VQA may not be of broad interest, which could limit its appeal and applicability to a wider audience.

The paper introduces the AssistGaze model but does not extensively compare it with existing models or approaches in the VQA domain. Such comparisons would have provided a clearer understanding of where AssistGaze stands in the current research landscape.

Given the novel nature of the dataset and the specific contexts in which it was collected, there might be concerns about the generalizability of the findings and the model to other domains or tasks beyond the ones explored in the paper.

The paper might lack an in-depth technical exploration of how the AssistGaze model effectively grounds perceptual input to semantic information.

**Reproducibility:**

2: Would be hard pressed to reproduce the results. The contribution depends on data that are simply not available outside the author's institution or consortium; not enough details are provided.

**Reviewer Confidence:**

3: Pretty sure, but there's a chance I missed something. Although I have a good feel for this area in general, I did not carefully check the paper's details, e.g., the math, experimental design, or novelty.

---

> ### Author Rebuttal · Authors · 2023-08-29
>
> ## Rebuttal for Reviewer 3c3r
>
> I would like to clarify the concerns of the reviewer in four aspects that are diversity, of the dataset, applicability and generalizability of the dataset, comparison of our model with existing approaches, and utilization of gaze information.
>
> The GazeVQA dataset creates a new sub-area and opportunities for task-oriented collaborative applications by combining two high-level research topics. Thus, the communities could do unique research to find solutions to the collaborative studies with instructional video by using new video analytics multi-modal approaches.
>
> ---
>
> ### **W1: Diversity of the dataset**
>
> * __Existing task-oriented datasets usually focus on specific scenarios.__
>     * Unlike kitchen-based niche tasks like EPIC-KITCHEN [1], and the Breakfast Actions Dataset [2], __GazeVQA is focused on task-oriented collaborative applications__.
>     * It is unclear how models which are trained with products such as toys that may be difficult to adapt to real-world applications, such as MECCANO [3] and Assembly101 [4], will perform in studies with real-world products.
>     * Although the sample number of 22 participants raises concerns, GazeVQA mitigates this concern with __125 hours of video data from 3 different video angles, more than 40 components, and more than 40 action steps that are not symmetrical__.
>
> ---
>
> ### **W2: Applicability and Generalizability of the dataset**
>
> * __Assembly/disassembly is a representative process in industry tasks and covers many important scenarios__.
>     * In our experiment, while the operation processes are similar across subjects, there are __many variations including the layout of the components, different styles of users completing a step, and collaboration interaction between instructor and user__. As such, we hope to strike a good balance of procedure consistency and variations, which is important in industrial operations.
>
> * __Successful contribution of gaze information__
>     * Tables 2,3,4,5 and 6 show that gaze information contributes to the model making correct predictions __when the gaze information is used in addition to other features__.
>     * Besides providing semantic information for industrial task assistance studies of gaze information, __it can be used as valuable input for multi-modal models in procedural planning with instructional videos, and human-centered user modeling studies__.
>
> * __Broad Review for Applicability and Generalizability of the gaze-enhanced dataset__
>     * __Gaze information plays a role as a potential contributor to studies in multi-disciplinary areas__  such as psychology, computer vision, NLP, and many more. For instance,
>         * Estimation of users' intentions and final decisions [5],
>         * Face-to-face communications in augmented reality environments [6],
>         * Users' manipulation of objects in virtual environments [7],
>         * Exploring the interaction of gaze, image, and users' description [8],
>         * Statistical word learning using eye movements [9],
>         * Analyzing the self-corrections for image descriptions with the help of eye-tracking corpus [10],
>         * Inspection of the images via spoken descriptions and gaze measurements [11].
>
> * In the context of task-oriented collaborative studies, GazeVQA can be generalized for numerous collaborative tasks such as __assembly-disassembly tasks__, __task-assisted video question-answering (VQA)__, __human decision-making__, __intention prediction research__, and __AR/VR studies__ by using the AssistGaze model strategy that utilizes more than twenty-five thousand generated questions and leverages gaze information for multimodal answer predictions.
> ---
>
> ### **W3: Comparison with existing approaches**
>
> * __AssistGaze structure is different from existing approaches__
>     * Unlike models that focus on one type of answer prediction, __AssistGaze is capable of multimodal input and three different answer predictions__.
> * __Baselines of GazeVQA__
>     * __Promising results are proposed with our baselines to follow the gaze-enhanced models for task-oriented collaborative industrial applications__.
>
>     * The recent SOTA model “VIOLET” [16] (100+ citations to date) which has only textual answer type, is used for benchmarking. VIOLET uses a Video Swin Transformer [17] to encode the images and BERT [18] to encode the text.
>     * Due to the limited time period for rebuttal, the VIOLET [16] model with its pertained weight with 1 epoch was fine-tuned. __We intend to provide updated results during the designated discussion period__, once model convergence through the fine-tuning process. __Results are shown in the Table below__.
>
> | Methods        |Answering Type | mAP                    |
> | -------------  | ------------- |-------------           |
> | Random Guess   |Text           | 0.07117 (from paper)   |
> | VIOLET [5]     |Text           | 0.09396 (to be updated)|
> | Ours           | Text          | 0.27173 (from paper)   |
> | Ours           | Text + Video + Image | 0.67199 (from paper) |
>
> * __We only fine-tuned the classification head of the VIOLET [16] model, while keeping the parameters of Swin Transformer [17] and BERT [18] frozen due to the time and computation resources limit__.  Hence the VIOLET model is underperforming compared to our results.
>
> * According to the experiments and ablation studies done for the study, we have found the reason for performance differences between 8 and 16 frames of CLIP. The reason 8 Frames of CLIP Features achieve better performance than 16 Frames of CLIP Features is that __the 16-frame version doubles the sequence length, making it relatively difficult to converge within 20 epochs__. In addition, 16 frames per video means more information is taken into consideration when the model is performing classification tasks. __There could be more noise in the features of 16 frames which results in the underperformance of the model__.
>
> ___
>
> ### **W4: Utilization of gaze information**
>
> * __Gaze information for action prediction__
>     * Gaze information has been investigated in the literature as gaze prediction [12].
>     * Action recognition studies can be performed by estimating gaze information using egocentric perspectives [13].
>     * It is also common to use head/hand motion and hand location/pose [14] or to use sensors in virtual environments [15] to estimate gaze location information.
>
>
> * __Gaze information for intention prediction__
>     * When processing a video of an action from an egocentric perspective, predicting the user's next steps based on various surrounding objects can hinder effective and precise answer predictions. However, __collecting the user's specific gaze location information enables user-object interactions and facilitates more accurate predictions of the user's intentions in the subsequent steps__.
>     * __The results of the experiments conducted in this study (shown in Fig. 1) illustrate that with the gaze information used in addition to other features, the model can yield more precise answer predictions__.
>
> Gaze information is not overly reliant on a particular kind of modalities, but it also has crucial contributions in many different research areas and application scenarios. Consequently, differing from recent studies, the utilization of gaze information for textual, image, and video answers in VQA and task-assistance research further highlights the contribution we present to the communities.
>
> ---
> ### **REFERENCES**
> 1. Damen, D., Doughty, H., Farinella, G. M., Furnari, A., Kazakos, E., Ma, J., ... & Wray, M. (2022). Rescaling egocentric vision: Collection, pipeline and challenges for epic-kitchens-100. International Journal of Computer Vision, 1-23.
> 2. Kuehne, H., Arslan, A., & Serre, T. (2014). The language of actions: Recovering the syntax and semantics of goal-directed human activities. In Proceedings of the IEEE conference on computer vision and pattern recognition (pp. 780-787).
> 3. Ragusa, F., Furnari, A., Livatino, S., & Farinella, G. M. (2021). The meccano dataset: Understanding human-object interactions from egocentric videos in an industrial-like domain. In Proceedings of the IEEE/CVF Winter Conference on Applications of Computer Vision (pp. 1569-1578).
> 4. Sener, F., Chatterjee, D., Shelepov, D., He, K., Singhania, D., Wang, R., & Yao, A. (2022). Assembly101: A large-scale multi-view video dataset for understanding procedural activities. In Proceedings of the IEEE/CVF Conference on Computer Vision and Pattern Recognition (pp. 21096-21106).
> 5. Huang, C. M., Andrist, S., Sauppé, A., & Mutlu, B. (2015). Using gaze patterns to predict task intent in collaboration. Frontiers in psychology, 6, 1049.
> 6. Rivu, R., Abdrabou, Y., Pfeuffer, K., Esteves, A., Meitner, S., & Alt, F. (2020, June). Stare: gaze-assisted face-to-face communication in augmented reality. In ACM Symposium on Eye Tracking Research and Applications (pp. 1-5).
> 7. Pathmanathan, N., Becher, M., Rodrigues, N., Reina, G., Ertl, T., Weiskopf, D., & Sedlmair, M. (2020, June). Eye vs. head: Comparing gaze methods for interaction in augmented reality. In ACM Symposium on Eye Tracking Research and Applications (pp. 1-5).
> 8. Yun, K., Peng, Y., Samaras, D., Zelinsky, G. J., & Berg, T. L. (2013). Studying relationships between human gaze, description, and computer vision. In Proceedings of the IEEE Conference on Computer Vision and Pattern Recognition (pp. 739-746).
> 9. Yu, C., & Smith, L. B. (2011). What you learn is what you see: using eye movements to study infant cross‐situational word learning. Developmental science, 14(2), 165-180.
> 10. Van Miltenburg, E., Kádár, A., Koolen, R., & Krahmer, E. (2018, August). DIDEC: The Dutch image description and eye-tracking corpus. In Proceedings of the 27th International Conference on Computational Linguistics (pp. 3658-3669).
> 11. Vaidyanathan, P., Prud’Hommeaux, E., Pelz, J. B., & Alm, C. O. (2018, July). SNAG: Spoken narratives and gaze dataset. In Proceedings of the 56th Annual Meeting of the Association for Computational Linguistics (Volume 2: Short Papers) (pp. 132-137).
> 12. Li, Y., Liu, M., & Rehg, J. (2021). In the eye of the beholder: Gaze and actions in first person video. IEEE transactions on pattern analysis and machine intelligence.
> 13. Min, K., & Corso, J. J. (2021). Integrating human gaze into attention for egocentric activity recognition. In Proceedings of the IEEE/CVF Winter Conference on Applications of Computer Vision (pp. 1069-1078).
> 14. Li, Y., Fathi, A., & Rehg, J. M. (2013). Learning to predict gaze in egocentric video. In Proceedings of the IEEE international conference on computer vision (pp. 3216-3223).
> 15. Emery, K. J., Zannoli, M., Xiao, L., Warren, J., & Talathi, S. S. (2021, March). Estimating Gaze From Head and Hand Pose and Scene Images for Open-Ended Exploration in VR Environments. In 2021 IEEE Conference on Virtual Reality and 3D User Interfaces Abstracts and Workshops (VRW) (pp. 554-555). IEEE.
> 16. Fu, T. J., Li, L., Gan, Z., Lin, K., Wang, W. Y., Wang, L., & Liu, Z. (2021). Violet: End-to-end video-language transformers with masked visual-token modeling. arXiv preprint arXiv:2111.12681.
> 17. Liu, Z., Ning, J., Cao, Y., Wei, Y., Zhang, Z., Lin, S., & Hu, H. (2022). Video swin transformer. In Proceedings of the IEEE/CVF conference on computer vision and pattern recognition (pp. 3202-3211).
> 18. Devlin, J., Chang, M. W., Lee, K., & Toutanova, K. (2018). Bert: Pre-training of deep bidirectional transformers for language understanding. arXiv preprint arXiv:1810.04805.

---

### Official Review · Reviewer_Ab7b · 2023-08-04

**Soundness:** 4

**Excitement:**

4: Strong: This paper deepens the understanding of some phenomenon or lowers the barriers to an existing research direction.

**Paper Topic And Main Contributions:**

**Topic**: This paper aims to provide a task-oriented collaborative QA application with eye gaze information, and builds an FPV video dataset(including TPV and multiple-choice QA pairs) and model.


**Contributions:**
1. a new VQA dataset(GazeVQA) with gaze information, which is composed of the assembly task with FPV and TPV videos.
2. a pipeline (or a model) for the implementation of the Gaze VQA

**Reasons To Accept:**

1. The paper provides a new Gaze VQA dataset, which can promote robot-human interaction and collaboration studies.
2. It implements a Gaze VQA pipeline and provides some ablations to show the effectiveness of different components.

**Reasons To Reject:**

1. The scenarios of the dataset are limited and only contain assembly and disassembly.
2. The details of the dataset are insufficient, such as the number(type) of the assembled objects/steps
3. The paper lacks comparisons (results on the current dataset) with existing frameworks, like VideoQA and ImageQA.

**Reproducibility:**

4: Could mostly reproduce the results, but there may be some variation because of sample variance or minor variations in their interpretation of the protocol or method.

**Reviewer Confidence:**

5: Positive that my evaluation is correct. I read the paper very carefully and I am very familiar with related work.

---

> ### Author Rebuttal · Authors · 2023-08-29
>
> ## Rebuttal for Reviewer Ab7b
>
> The GazeVQA dataset focuses on task-oriented collaborative applications such as assembly-disassembly to show that there is a strong link between VQA studies and instructional videos.
>
> ---
>
> ### **W1: Scenarios of the dataset**
>
> * __Assembly-disassembly tasks are the fundamental scenarios which are commonly realized in every kind of industrial-based application__.
>     * These tasks cover instructional procedures. Therefore, GazeVQA represents better approaches in different areas such as human-robot interaction, and task assistance by conducting these experiments with the collaborative version of these tasks.
>     * Our question set is prepared to be adapted to different industrial assembly-disassembly datasets by completing the name of object adjustments.
>     * These explain why assembly disassembly constitutes a valid process that is generic enough to support broad industrial scenarios/tasks.
>
> * __Existing task-oriented datasets usually focus on specific scenarios.__
>     * While EPIC-KITCHEN [1] and the Breakfast Actions Dataset [2], which are widely adopted in the literature, focus on __a limited number of niche kitchen scenarios__, MECCANO [3], and Assembly101 [4] also focus on toy-based assembly-disassembly scenarios.
>
> * __Diversity of products, scenarios, and questions__
>     * Instead of using a small-sized toy as used in Asembly101 [4] and MECCANO [5] studies, __GazeVQA uses the 3D-printed, and one-to-one scale of the gearbox__ in the experiments.
>     * The assembly and disassembly processes are __designed based on real practices in the industry__, which makes them of practical value and challenging for visual analytics approaches.
>     * The usage of “a real-world industrial product, more than 25K adaptable questions, more than 40 components and scenarios in the dataset” increases GazeVQA's __generalizability, applicability, and adaptability to different real-world applications__ compared to current studies.
>
> * __New opportunities__
>     * GazeVQA has unique characteristics that are not available in EPIC-KITCHEN [1], Breakfast Actions Dataset [2], MECCANO [3], and Assembly101 [4] datasets.
>     * Focusing on specific improvements in developing areas could be more beneficial for model design and training.
>     * GazeVQA propose __novel components and scenarios for task-oriented studies__.
>     * GazeVQA __offers a new combined subfield to the communities__ for new research contributions.
>
> ---
>
> ### **W2: Details of the dataset**
>
> * __Numbers and types of the components__
>     * Maximum of 22 steps in the assembly task and 19 steps in the disassembly task.
>     * Assembling is started with the instruction which is “___Place the casing top on the pallet. Exterior facing up___ and over with the instruction which is “ ___Put the gearbox horizontally___ ”.
>     * Disassembling is started with the instruction which is “___Put the gearbox in a vertical position.___ ” and over with the instruction “___Unload the casing base___ ”.
>     * There are 1 casing top, 1 casing base, 2 output hub cover, 1 input hub covers, 3 small hub covers, 22 hub cover bolts, 3 sub-assemblies (1 input shaft, 1 transfer shaft, and 1 output shaft), 6 casing bolts, 2 oil level indicators, and 1 breather plug. __In total, 42 components are used to assemble or disassemble this product.__
>     * __Components and steps are illustrated with relevant instructions as shown in Figures 10 and 11.__
>     * We will include detailed procedures with visualization in the supplementary material.
>
> ---
>
> ### **W3: Comparison with existing frameworks**
>
> * __Different model structure from existing methods__
>     * AssistGaze predicts 3 different types of answers by using multimodal inputs, which __has not been covered in detail in the literature__.
> * __Fair comparison__
>     * The most effective result is achieved when all features are used together.
>     * __Due to significant differences such as input-output types, modalities, and the diversity of the numbers and types of output predictions__, a fair and direct comparison between these studies could be infeasible.
>     * We compared our model with the recent SOTA model “VIOLET” [5] (100+ citations to date) whose answer type only has textual answers. VIOLET uses a Video Swin Transformer [6] to encode the images and BERT [7] to encode the text.
>     * Due to time constraints, we fine-tuned the VIOLET [5] model with its pertained weight with 1 epoch. __We intend to provide updated results during the designated discussion period__, once model convergence through the fine-tuning process. __Table below shows the results__.
>
> | Methods        |Answering Type | mAP                    |
> | -------------  | ------------- |-------------           |
> | Random Guess   |Text           | 0.07117 (from paper)   |
> | VIOLET [5]     |Text           | 0.09396 (to be updated)|
> | Ours           | Text          | 0.27173 (from paper)   |
> | Ours           | Text + Video + Image | 0.67199 (from paper) |
>
> We only fine-tuned the classification head of the VIOLET [5] model, while keeping the parameters of Swin Transformer [6] and BERT [7] frozen due to the time and computation resources limit.  Hence the VIOLET model is underperforming compared to our results.
>
> ---
>
>
> ### **REFERENCES**
>
> 1. Damen, D., Doughty, H., Farinella, G. M., Furnari, A., Kazakos, E., Ma, J., ... & Wray, M. (2022). Rescaling egocentric vision: Collection, pipeline and challenges for epic-kitchens-100. International Journal of Computer Vision, 1-23.
> 2. Kuehne, H., Arslan, A., & Serre, T. (2014). The language of actions: Recovering the syntax and semantics of goal-directed human activities. In Proceedings of the IEEE conference on computer vision and pattern recognition (pp. 780-787).
> 3. Ragusa, F., Furnari, A., Livatino, S., & Farinella, G. M. (2021). The meccano dataset: Understanding human-object interactions from egocentric videos in an industrial-like domain. In Proceedings of the IEEE/CVF Winter Conference on Applications of Computer Vision (pp. 1569-1578).
> 4. Sener, F., Chatterjee, D., Shelepov, D., He, K., Singhania, D., Wang, R., & Yao, A. (2022). Assembly101: A large-scale multi-view video dataset for understanding procedural activities. In Proceedings of the IEEE/CVF Conference on Computer Vision and Pattern Recognition (pp. 21096-21106).
> 5. Fu, T. J., Li, L., Gan, Z., Lin, K., Wang, W. Y., Wang, L., & Liu, Z. (2021). Violet: End-to-end video-language transformers with masked visual-token modeling. arXiv preprint arXiv:2111.12681.
> 6. Liu, Z., Ning, J., Cao, Y., Wei, Y., Zhang, Z., Lin, S., & Hu, H. (2022). Video swin transformer. In Proceedings of the IEEE/CVF conference on computer vision and pattern recognition (pp. 3202-3211).
> 7. Devlin, J., Chang, M. W., Lee, K., & Toutanova, K. (2018). Bert: Pre-training of deep bidirectional transformers for language understanding. arXiv preprint arXiv:1810.04805.

---

### Meta-Review · Area_Chair_9wVj · 2023-09-18

**Recommendation:** 4

**Metareview:**

The paper has received positive feedbacks from all reviewers. All reviewers agree the dataset is a unique and important contribution to robot-human interaction and collaboration studies, while there have been concerns over dataset generalization and a lack of experimental comparison with other VQA methods, the authors provided satisfactory responses leading reviewers to increase their scores.

---

### Decision · Program_Chairs · 2023-10-07

**Decision:**

Accept-Main

**Comment:**

The paper has received positive feedbacks from all reviewers. All reviewers agree the dataset is a unique and important contribution to robot-human interaction and collaboration studies, while there have been concerns over dataset generalization and a lack of experimental comparison with other VQA methods, the authors provided satisfactory responses leading reviewers to increase their scores.